# When Autoantibodies Are Missing: The Challenge of Seronegative Rheumatoid Arthritis

**DOI:** 10.3390/antib12040069

**Published:** 2023-10-31

**Authors:** Marino Paroli, Maria Isabella Sirinian

**Affiliations:** Center for Allergy and Immunology, Department of Clinical, Internal, Anesthesiologic and Cardiovascular Sciences, Sapienza University of Rome c/o Polo Pontino, 04100 Latina, Italy

**Keywords:** seronegative rheumatoid arthritis, rheumatoid factor, anti-citrullinated protein antibodies

## Abstract

Seronegative rheumatoid arthritis (SNRA) is characterized by the absence of both rheumatoid factor (RF) and antibodies against the cyclic citrullinated protein (ACPA) in serum. However, the differences between the two forms of RA are more complex and have not yet been definitively characterized. Several lines of evidences support the idea that there are specific elements of the two forms, including genetic background, epidemiology, pathogenesis, severity of progression over time, and response to therapy. Clinical features that may differentiate SNRA from SPRA are also suggested by data obtained from classical radiology and newer imaging techniques. Although new evidence seems to provide additional help in differentiating the two forms of RA, their distinguishing features remain largely elusive. It should also be emphasized that the distinctive features of RA forms, if not properly recognized, can lead to the underdiagnosis of SNRA, potentially missing the period called the “window of opportunity” that is critical for early diagnosis, timely treatment, and better prognosis. This review aims to summarize the data provided in the scientific literature with the goal of helping clinicians diagnose SNRA as accurately as possible, with emphasis on the most recent findings available.

## 1. Introduction

Rheumatoid arthritis (RA) is an inflammatory disease that primarily affects synovial joints through an autoimmune mechanism. If not treated properly, the disease can lead to bone erosion, joint deformities, and disability. Arthritis can also cause serious extra-articular disorders, including interstitial lung disease, vasculitis, and lymphoma [1,2]. According to the latest 2010 ACR/EULAR criteria [3], the diagnosis is based on a scoring system calculated using symptom duration, the number and type of joints affected, altered acute-phase reactants, and the presence of autoantibodies, such as rheumatoid factor (RF) and/or anti-citrullinated protein antibodies (ACPAs), in serum [4]. Because the presence of RF and ACPA in serum is not necessary for the diagnosis of RA, a substantial number of patients presenting with the typical clinical features of RA in the absence of these autoantibodies can be diagnosed as having RA. The form of RA without RF and/or ACPA is termed seronegative RA (SNRA) [5,6]. Numerous observations have reported that the clinical presentation, course severity, and response to therapy appear to be significantly different between SNRA and seropositive RA (SPRA) [7,8,9]. In recent years, the focus on seronegative forms of RA has increased due to clinicians’ sensitivity to the different clinical presentations of RA, as well as the advent of increasingly sophisticated means of both molecular and imaging investigations. In addition, the availability of therapeutic means that can act on different effector functions of immunity has indirectly clarified further distinctive aspects between SNRA and SPRA. The purpose of this review is to summarize the distinct elements that have emerged over time regarding the epidemiological, pathogenetic, and clinical features that help to distinguish SNRA from SPRA.

## 2. Epidemiology

Available epidemiological data have traditionally reported a lower prevalence of SNRA than SPRA, ranging from 20 to 30 percent of total cases of RA [10,11]. However, the incidence of SNRA has been reported to be increased in recent decades [12,13]. Many hypotheses have been advanced to explain this finding. One possible cause is the increasing age of the general population. In fact, late-onset RA occurring in elderly patients is commonly seronegative, suggesting that the dysregulation of inflammation, typical in people of old age, may underlie SNRA [14,15,16]. Another cause is an overall reduction in smoking habits, with cigarette smoking being a strong risk factor for protein citrullination [17]. It is believed that the process of citrullination, by changing the self-nature of joint antigens to non-self-antigens, induces an autoimmune process that leads to the generation of ACPA and causes the humoral and cellular immune systems to attack the altered joint antigens, resulting in synovitis with tissue damage. Therefore, it is likely that the reduction in smoking increased the incidence ratio between SNRA and the seropositive forms through this immunological mechanism [18]. Other factors reported to explain the increased incidence/prevalence of SNRA are changes in the microbiome, possibly from chronic triggers by gut flora via microbial DNA and pepdidoglycans [19], and some environmental factors, including increased occupational exposure to crystalline free silica [20]. Of course, further studies are needed to clarify whether the upward trend in the incidence/prevalence of SNRA compared with that of SNPA is a finding that can proceed over time and whether there are additional genetic and/or environmental factors causing it, which have yet to be elucidated.

## 3. Pathogenesis

The first distinction between SNRA and SPRA is their different genetic backgrounds. Although RA is a polygenic disease, some genetic risk factors have been identified for SNRA. Among them, the HLA-B*08/DRB1*03 haplotype is one of the genetic markers most frequently associated with SNRA, while the classical HLA-DRB1*04 and *10 alleles have been shown to be risk factors exclusively for SPRA [21,22]. Non-HLA genes also play a relevant role in determining susceptibility to RA, including mutations in genes coding for Janus kinase (JAK)/signal transducer and activator of transcription (STAT) proteins, which are risk factors for SPRA but not for SNRA [23,24]. A genome-wide association study revealed an association with single-nucleotide polymorphism of non-HLA genes *ANKRD55* [25] and *CLYBL* [26] in SNRA but not in SPRA. It is possible to speculate that mutations in non-HLA genes represent changes in the innate-type immune response through the modulation of the synthesis of cytokines and other soluble factors. Indeed, innate immunity seems to be more prominent in SNRA than in SPRA, while antigen/autoantigen presentation to T lymphocytes by HLA molecules represents a key element of adaptive immunity that is more typical of SPRA pathogenesis.

The study of the expression of miRNA is a new exciting field of research aiming to identify biomarkers for differentiating SNRA from SPRA. In this regard, it was recently reported that the miRNAs has-miR-362-5p and has-miR-708-3p were upregulated in SNRA but not in SPRA. Other miRNAs were found to be downregulated differently in the two forms of RA, including the mRNAs expressed exclusively in SPRA and others common to both forms [27,28]. Table 1 summarizes the different genetic backgrounds and miRNA expressions between SNRA and SPRA.

Some studies were conducted to determine the differences between SPRA and SNRA at the cellular level. The synovial histological score for CD4+ T cells, CD68+ cells in the lining layer, and sublining CD3+ and vessel CD31+ positive cells was less abundant in undifferentiated seronegative arthritis than in differentiated SPRA [28]. It has also been reported that synovium-infiltrating monocytes and macrophages predominate in SNRA [21]. In an attempt to identify biomarkers that can differentiate SNRA from psoriatic arthritis (PsA) because they share some clinical features, a study was conducted that analyzed the synovial histopathology of the two diseases. It was reported that plasma cells predominate in the synovium of SNRA, while mast cells predominate in PsA [31].

An immunohistochemical analysis of the synovium also revealed a higher percentage of tissue-resident dendritic cells and a reduced expression of the PD-1 inhibitory receptor on T cells in SNRA compared with its seropositive counterpart [32]. Therefore, the finding that the immune checkpoint inhibitor PD-1 can induce SNRA in the course of cancer therapy is of particular interest [33]. This observation is discussed in more detail in a later section of this review for its potential therapeutic implications. Table 2 shows the inflammatory cells detected in the synovial membrane during SNRA and SPRA.

Interestingly, SNRA occurrence has been reported during asthma therapy with anti-IL4/IL-13 biologics with the activation of the IL-23-IL-17 axis, suggesting a protective role of T helper-2 (TH-2) cells in the disease [34]. This evidence further supports the idea that SNRA is a form of RA that diverges substantially from SPRA and suggests a similarity of SNRA with SpA, which depends primarily on IL-17 [5,35,36]. Several observations point out that SNRA has a more variable outcome, generally associated with a better prognosis than SPRA [37,38]. It is interesting to note the reported association between SNRA with NLRP3 inflammasome activation. In this regard, studies have demonstrated a role for interleukin-beta (IL1β), a key component of the inflammasome, in the pathogenesis of SNRA [39]. The pathogenetic relationship between SNRA and IL-1β may explain the favorable response to the interleukin-1 receptor antagonist (IL-1ra), as observed in some patients with SNRA, and the minor response to JAK inhibitor (JAKi) therapy of SNRA compared with SPRA, as reported in some studies [40,41,42]. This can be related to the fact that IL-1 does not depend on the JAK/STAT transduction pathway. As is well known, the activation of the NLRP3 inflammasome by monosodium urate crystals with the release of IL-1β plays a major role in the initiation of gout flare [43]. Interestingly, elevated uric acid levels and crystal deposition are occasionally observed in SNRA but not in SPRA, indirectly suggesting an at least partially autoinflammatory nature of SNRA [43]. Although it is not easy to give an explanation for these observations, they suggest a possible pathogenetic link between SNRA and crystal deposition arthritis. Similarly, an autoinflammatory nature has also been proposed for spondyloarthritis (SpA) [44]. To elucidate the possible autoinflammatory component of SNRA, further studies using methods to study the inflammasome and the genetic substrate of this form of RA are needed. In addition, it should be noted that the study of synovial histology is providing very promising results due to the precise characterization of the cells that infiltrate this tissue. Using the methods of histochemistry and flow cytometry, many research groups are trying to identify new biomarkers that can differentiate SNRA from SPRA. Although synovial biopsy is an invasive procedure, it cannot be ruled out that, in the near future, the results obtained may allow for the development of serologic tests that allow for differential diagnosis through simpler diagnostic tests. Pathogenetic characterization, of course, not only has a scientific or diagnostic purpose but also appears essential for the setting of targeted therapies and the possible realization of the so-called personalized therapy tailored to the individual.

## 4. Diagnosis

As discussed earlier, the diagnosis of RA based on 2010 ACR/EULAR criteria is likely when the patient’s signs and symptoms reach a score of at least 6 [3]. However, this classification criterion may not be optimal for the diagnosis of SNRA. In fact, because RF and ACPA contribute significantly to this score, a seronegative disease must have a higher clinical and inflammatory severity than SPRA to compensate for the lack of serologic markers [45,46]. Although studies are underway to identify as-yet unknown autoantibodies, such as anti-modified protein antibodies (AMPAs), that could help in the diagnosis of SNRA, these efforts have so far been unsuccessful, and we currently have no biomarkers for this condition [47,48]. In a cohort study, it was shown that the diagnosis of SNRA was significantly delayed compared with that of SPRA, even when the previous 1987 RA diagnostic criteria were met [49]. Inadequate classification criteria could lead to a delay in the initiation of therapy in SNRA, resulting in the inability to take advantage of the so-called “window of opportunity” [50], which is critical for achieving early remission of the disease [51].

Therefore, it is likely that, based on the 2010 ACR/EULAR criteria, SNRA is underdiagnosed, suggesting that its incidence/prevalence is actually higher than that reported in the literature. The results of registry studies indicating a low incidence of SNRA may therefore be attributed, at least in part, to a missed diagnosis due to the new classification [52,53]. Thus, the clinical differences between SNRA and SPRA have been carefully analyzed in several studies to help physicians make a correct diagnosis, in addition to the official classification criteria. In initial studies, detectable differences with standard radiography were reported to distinguish possible differences between the two conditions at the hand level [54,55]. Notably, the van der Heijde-modified total Sharp score was significantly lower in patients with SNRA for both erosions and joint space narrowing than in those with SPRA. Interestingly, the erosion subscore in SNRA was higher in the carpal and proximal interphalangeal compartments and in the distal ulna, while the joints of the feet were virtually spared, with a greater involvement of the large and carpal joints in SNRA than in SPRA [56,57]. However, in SPRA, joint damage was significantly more evident in metacarpophalangeal joints II, III, and V, with bone demineralization [58]. In general, the involvement of the metacarpophalangeal joint appears to be a hallmark of SPRA, whereas it is more rarely found in SNRA [59].

To clarify the differences between SNRA and SPRA and improve the accuracy of diagnosis, other criteria have been proposed based on imaging techniques other than standard radiography. Musculoskeletal ultrasound (MSK-US) has emerged in recent decades as a useful and inexpensive tool for diagnosing RA. Several studies have pointed out that this method is very effective in detecting the presence of synovitis compared with clinical examination [60,61]. The usefulness of this tool was emphasized with the inclusion of this method as a means of determining the presence of synovitis in the 2010 ACR/EULAR diagnostic criteria for RA [3,62]. While for the diagnosis of SPRA, some studies have not found significant advantages of MSK-US over physical examination [63], this imaging tool has proven very useful in helping physicians diagnose SNRA. A low degree of synovitis and frequent tenosynovitis involvement, which are hallmarks of SNRA, can be detected with a high accuracy via MSK-US [64,65], as recently summarized in a systematic review of the literature [66]. In a recent study that exploited MSK-US for the evaluation of tendon involvement in SNRA, inflammation of the synovial sheaths of the extensor tendons of the carpus, particularly the sixth compartment, and of the finger flexors, particularly the third finger, was found to be a hallmark of the disease. The usefulness of microvascular flow imaging was also demonstrated in the same study [67]. The results obtained by MSK-US, indicating that the presence of tenosynovitis is more frequent in SNRA than in SPRA and that the extra synovial compartment is often involved in SNRA, were also confirmed by MRI investigation [68,69]. Therefore, there is broad agreement that MSK-US provides significant help in differentiating between SNRA and SPRA [70]. The most important clinical features that differentiate SNRA from SPRA are summarized in Table 3.

However, clinical similarities with SpA pose a risk of misdiagnosis, which could lead to an overestimation of the prevalence of SNRA [72]. Given the substantial differences between SNRA and SNRA, several studies have even questioned the actual existence of SNRA. SNRA presentation may show similarities with different types of SpAs, including PsA, axial SpA (axSpA), and undifferentiated SpA. In particular, PsA, as discussed earlier, has the most similar clinical features to SNRA [73]. With the refinement of methods and instruments, some typical features of SpA, such as enthesitis, can be detected with sufficient accuracy, allowing for a proper differential diagnosis [74,75]. At a more advanced stage, however, axial manifestations may appear, which may cause SNRA to be reclassified as axSpA [76].

It cannot be excluded that, in clinical practice, some rheumatic diseases diagnosed as SNRA are SpA with predominantly peripheral expression [71,77]. In some real-world studies, it has been reported that the initial diagnosis of SNRA has often been changed over time to that of SpA [77,78]. In a study with a 23-year follow-up, more than 60 percent of patients initially classified as having SNRA turned out to have SpA over time [79]. This aspect is of paramount importance in the clinical setting. Indeed, RA and SpA share some initial treatments, such as TNFi. However, after the eventual primary or secondary failure of these treatments, the drugs used in the two diseases are very different. In RA, anti-interleukin-6-receptor drugs or drugs that interfere with adaptive immunity are used, although the latter, as we discussed earlier, are not very effective in SNRA. The treatment of SpA, however, makes use of drugs that interfere with the IL-23/IL17 axis, which are ineffective in the two different forms of RA. A correct diagnosis is therefore crucial for the prescriptive appropriateness of the rheumatic patient and a satisfactory therapeutic outcome.

Differences in the comorbidities most frequently associated with SNRAs or SPRAs were investigated in a large study of data obtained from Swedish registries. Specifically, SNRA showed an increase in atrial fibrillation, type II diabetes, psychiatric disorders, neoplasms, and peripheral neuropathy compared with its seropositive counterpart [80]. In another study, cardiovascular risk was shown to be significantly higher in SPRA than in SNRA [81]. Finally, fibromyalgia was significantly associated with SNRA but not SPRA [82], although not all studies have confirmed this finding [83].

## 5. Severity of the Disease

An unresolved issue concerns the long-term prognosis of SNRA compared with SPRA, both in terms of joint damage and extra-articular manifestations [77]. Many prospective studies have underlined a better prognosis for SNRA than SPRA, even when presenting with a higher disease activity [84,85,86]. Moreover, the two forms of RA appear to be different diseases in some respects, as SNRA has distinct genetic characteristics and appears to develop under peculiar environmental conditions [87,88,89]. SNRA is generally considered a benign form of RA, and this has been attributed to the absence of autoantibodies in serum. In fact, several studies have shown that the erosive evolution of RA is associated with the presence of ACPA, regardless of disease activity or other clinical manifestations [90,91]. However, not all studies agree on the benign nature of SNRA [14]. It has been objected that studies demonstrating the greater severity of SNRA could be due to a bias arising from the current 2010 ACR/EULAR classification criteria. In fact, since these criteria assign a significant score to the presence of RF and/or ACPA, their absence in serum must be compensated for by increased joint involvement and inflammatory status to reach the score necessary for a diagnosis of probable RA [45,92], as discussed above. Prospective registry studies will be able to clarify whether SNRA actually represents a form of RA with a benign course or, conversely, whether its pathogenicity is currently underestimated.

## 6. The Response to Therapy of SNRA Is Different from That of SPRA

According to the ACR/EULAR recommendations, the initial therapy of RA should be based on the use of conventional synthetic antirheumatic drugs (csDMARDs) [93]. The first-line csDMARD suggested by most experts, regardless of serologic status, is methotrexate (MTX). However, several studies have shown that MTX is less effective in SNRA than in SPRA [94,95]. It has also been shown that MTX is not as effective in improving the retention rate of biologic drugs such as TNF inhibitors (TNFis) in SNRA compared with SPRA [96,97]. Since the enhanced action of biologics from combination therapy with MTX is attributed to the ability of MTX to inhibit the production of anti-drug antibodies (ADAs) by B lymphocytes [98], this evidence supports the hypothesis that adaptive immunity is less involved in SNRA than in SPRA. In support of this hypothesis is the observed low efficacy of drugs that target B or T cells, such as anti-CD20 or anti-CD80/CD86 biologics, respectively, in patients with SNRA compared with seropositive patients [99,100,101,102].

However, several lines of evidence suggest that SNRA is, in general, more responsive to therapy than SPRA, with a more favorable long-term radiographic outcome [79,84,85,103]. It is also noteworthy that SNRA, unlike SNRA, generally responds positively to switching to a second anti-TNFalpha after failure of the first and often does not require switching to other cytokine-targeted therapies [104]. Studies on the efficacy of Janus kinase inhibitors (JAKis) and their retention rates, which can inhibit the action of multiple pro-inflammatory cytokines in patients with SNRA, are not yet conclusive, and further studies are needed to clarify these important issues [41,105,106]. The results of several clinical trials and real-world data on response to therapy in SNRA and SPRA are summarized in Table 4. However, randomized clinical trials specific to SNRA have not yet been conducted. Therefore, this represents an important unmet need for the optimization of different clinical forms of RA.

As introduced in an earlier section of this review, interesting evidence regarding possible alternative therapies to SNRAs comes from the observation that inflammatory rheumatic diseases, termed immune-related rheumatologic adverse events (Rh-irAEs), can develop during cancer therapy with immune checkpoint inhibitors (ICIs) [107]. In particular, the occurrence of RA is particularly frequent and is seronegative in most cases [108]. In fact, it has been reported that RA appearing during ICI therapy actually resembles SNRA both clinically and serologically. In addition to the absence of RF and ACPA, morning stiffness and the absence of erosive joint damage were found to be present in the vast majority of patients with ICI-induced RA [109,110]. These findings indicate that the dysregulation of PD-1-mediated signaling could be at least partly responsible for the pathogenesis of SNRA via an antibody-independent mechanism, suggesting the use of PD-1 agonists as a potential new therapeutic option for RA. Encouraging results of a phase II trial on the use of the humanized monoclonal antibody peresolimab were recently published. However, although patients with SNRA and SPRA were included in the study, an analysis of the response of the two subgroups was not performed [111]. Taken together, all these results underscore that the response to SNRA therapy is a very complex issue. It should also be considered that the research and development of new drugs in immuno-rheumatology are particularly active. Therefore, it cannot be ruled out that new drugs that are more effective than the current ones may become available in the next few years and that their use may indirectly clarify the pathogenetic and clinical differences between SNRA and SPRA. Figure 1 summarizes the distinguishing features between SNRA and SNRA that have emerged so far.

## 7. Conclusions

SNRA represents an inflammatory rheumatic condition that is difficult to classify uniquely. Distinctive genetic features and the greater involvement of innate versus adaptive immunity represent characteristic features of this form of RA. This translates not only into distinctive immunopathologic features, such as less synovial involvement and a more frequent occurrence of tendinitis, but also into features of different prognoses and responses to therapy as compared with SPRA. Prospective and controlled studies are therefore needed to delineate an increasingly accurate picture of this form of RA, which could also allow for a better differentiation from SpA with which SNRA seems to share numerous pathogenetic and clinical features.

## Figures and Tables

**Figure 1 antibodies-12-00069-f001:**
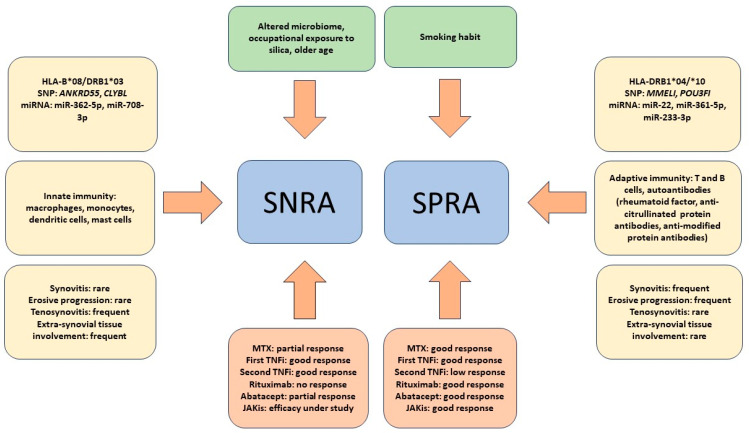
The figure summarizes the main factors described so far as being associated with seronegative rheumatoid arthritis (SNRA) or seropositive rheumatoid arthritis (SPRA). Epidemiological and environmental factors are depicted in the upper green quadrants. The yellow quadrants on either side of the figure summarize HLA and non-HLA genetic factors, microRNA (miR) expression, cells of immunity most involved in pathogenesis, and clinical features. The red quadrants at the bottom of the figure summarize the response to therapy with both conventional drugs, such as methotrexate (MTX), and biologic drugs, such as TNF inhibitors (TNFis), anti-CD20 (rituximab), and anti-CD80/86 (abatacept), as well as to Janus kinase inhibitors (JAKis).

**Table 1 antibodies-12-00069-t001:** Susceptibility to SNRA and SPRA is favored by the presence of HLA and non-HLA genetic factors and miRNA expression.

Feature	SNRA	SPRA	References
HLA-B*08/DRB1*03	High	Low	[21]
HLA-DRB1*04/*10	No	High	[29]
SNP *ANKRD55*	High	No	[25]
SNP *CLYBL*	High	No	[26]
has-miR-362-5p	High	Low	[27]
has-miR-708-3p	High	Low	[28]
has-miR-24	High	High	[30]
has-miR-125a	High	High	[30]
has-miR-22	Low	High	[30]
has-miR-361-5p	Low	High	[30]
has-miR-233-3p	Low	High	[30]

SNRA = seronegative rheumatoid arthritis; SPRA = seropositive rheumatoid arthritis; miRNA = microRNA; SNP = single-nucleotide polymorphism.

**Table 2 antibodies-12-00069-t002:** Immunological features of SNRA and SPRA.

Cell Type	SNRA	SPRA	References
Synovial lining CD68+ cells	Rare	Frequent	[28]
Synovial CD4+ T cells	Rare	Frequent	[28]
Synovial vessel CD31+ cells	Rare	Frequent	[28]
Synovial monocytes	Frequent	Rare	[21]
Synovial macrophages	Frequent	Rare	[21]
Plasma cells	Frequent	Rare	[32]
PD-1+ T cells	Rare	Frequent	[33]

SNRA = seronegative rheumatoid arthritis; SPRA = seropositive rheumatoid arthritis.

**Table 3 antibodies-12-00069-t003:** Typical clinical involvement of SNRA and SPRA.

Feature	SNRA	SPRA	References
mTSS	Low	High	[71]
MCP II damage	Low	High	[60]
MCP III damage	Low	High	[60]
MCP V damage	Low	High	[60]
Synovitis	Rare	Frequent	[66]
Tenosynovitis	Frequent	Rare	[67]
Extra-synovial involvement	Frequent	Rare	[70]

SNRA = seronegative rheumatoid arthritis; SPRA = seropositive rheumatoid arthritis; mTSS = van der Heijde modified total Sharp score; MCP = metacarpophalangeal joint.

**Table 4 antibodies-12-00069-t004:** Response to therapy of SNRA and SPRA with conventional and advanced drugs.

Drug	SNRA	SPRA	References
MTX	Limited	Good	[96]
TNFi	Good	Good	[104]
Second TNFi	Good	Limited	[104]
Anti-CD20 biologic	No	Good	[101]
Anti-CD80/CD86 biologic	Limited	Good	[102]
JAKis	Uncertain	Good	[105]

SNRA = seronegative rheumatoid arthritis; SPRA = seropositive rheumatoid arthritis; MTX = methotrexate; TNFi = anti-tumor necrosis factor-α biologic; JAKis = Janus kinase inhibitors.

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
