# Peer review of "When Autoantibodies Are Missing: The Challenge of Seronegative Rheumatoid Arthritis"

_2073-4468, 2023, doi:10.3390/antib12040069_

Round 1

Reviewer 1 Report

The Review was well written with detailed explanation of SNRA and SPRA. Below are the comments.

1.  Does miRNA's expression is seen in SPRA?

2. Page 5 : Under "Response to therapy".. it should be "unlike SPRA"

3. What are "+/-" in table 2, 3, and 4 

4. Does SNRA patients respond to "Peresolimab" ?

5. page 1: How reducing overall smoking causes SNRA?

6.  page 2: does change in microbiome cause SNRA?

7. page 2: "remove "presence of " in pathogenesis section 

8. page 2: Remove "a very comprehensive review......"

9. Page 2: any  miRNA expression related to SPRA?

10. Table 1: should be HLA-B08/DRB1*03

11. page 2: Remove "some of the available data on the infiltration of"

12. Table 2: what does "+/-"  represent?

13. A table showing the current available treatment and drugs in clinical trials for SNRA is highly important.

14. For this review, authors used 112 references, which are difficult to read. Does "Antibodies" journal have a limit for this?

Author Response

Responses to Reviewer #1

(Text changes related to responses to comments have been highlighted in yellow)

  1. Does miRNA's expression is seen in SPRA? Answer: miRNA expression is also seen in SPRA. However, the expression between the two forms varies. Some miRNAs are expressed exclusively in SNRA, some in SNRP, and some in both forms of RA. We have specified this evidence in the text and also added miRNAs expressed both by SNRP and uniquely in SPRA in Table 1.
  2. Page 5: Under "Response to therapy" it should be "unlike SPRA" Answer: The chapter title has been changed according to the suggestion for a better understanding of the content of the text.
  3. What are "+/-" in table 2, 3, and 4 Answer: The + and - symbols have been replaced with more understandable definitions.
  4. Does SNRA patients respond to "Peresolimab"? Answer: It was specified in the text that both patients with SNRA and SPRA were included in the study However, a comparison analysis between these two groups of patients was not performed.

  5. page 1: How reducing overall smoking causes SNRA? Answer: The hypothesis was that by reducing smoking habit, the percentage of patients with SNRA was relatively reduced due to the effect of smoking on protein citrullination. This concept has been added to the text.
  6. page 2: does change in microbiome cause SNRA? Answer: The most widely accepted hypothesis on the pro-inflammatory effect of bacterial DNA as well as other factors in their wall has been included in the text.

  7. page 2: "remove "presence of " in pathogenesis section Answer: The sentence has been removed

    8. page 2: Remove "a very comprehensive review......" Answer: The sentence has been removed

    9. Page 2: any miRNA expression related to SPRA? Answer: This information as mentioned above was included in Table 1.

    10. Table 1: should be HLA-B08/DRB1*03 Answer: The writing of HLA-B8 has been corrected as indicated.

    11. page 2: Remove "some of the available data on the infiltration of" Answer: The sentence has been removed

    12. Table 2: what does "+/-"  represent? Answer: As in Table 1, the + and - symbols have been replaced with more understandable definitions.
  8. A table showing the current available treatment and drugs in clinical trials for SNRA is highly important. Answer: We completely agree with the reviewer's observation. However, the data for the subgroup patients with SNRP were generally not analyzed by secondary analysis. Exclusive studies of patients with SNRA to our knowledge have not been conducted. Therefore, we supplemented Table 4 (response to therapy) with bibliographic entries on the most relevant clinical studies on the topic.

    14. For this review, authors used 112 references, which are difficult to read. Does "Antibodies" journal have a limit for this? Answer: Some redundant references have been removed. However, we checked with the editorial board, and according to the journal's policy it is appropriate for a review to have at least 100 bibliographic entries.

Reviewer 2 Report

Paroni and Sirinian have reviewed the less prevalent and less severe nonetheless less clear seronegative aspects of rheumatoid arthritis (SNRA) than its seropositive counterpart (SPRA). The authors highlighted on the knowledge base of genetic and immune risk factors for pathogenesis, current diagnosis and therapeutic aspects of SNRA. The key insights include dysregulation of inflammatory signaling pathway (e.g., PD-1) as a possible mechanism leading to SNRA, miRNA expression and SNP in genes as possible molecular biomarkers, and the need to reassess the current RA clinical diagnosis criteria to classify SNRA. The topics of discussion in this review are merited however some clarification into the authors’ insights is needed.

1. Please summarize in a table the distinctive features of SNRA and SPRA including but not limited to epidemiology, clinical features, biomarkers, diagnosis, therapeutic and prognostic features.

2. Table 1: It is not clear what positive and negative mean in the context since as I understand the results are based on miRNA/mRNA expression data. Please modify the table to clarify the fold change ratio in the expression. Also, add the references to those experimental findings in the table.

3. Pathogenesis –

paragraph 1: It is interesting to note the association of non-HLA genes with SNRA but not SPRA. Could the authors also highlight the functional role of the genes (and if any other) within the context of RA?

paragraph 5: If SpA is mentioned here for the first time, please write the full form.

4. It would be easier for a layperson reader to follow the contributing factors to the onset and progression of SNRA and SPRA based on available knowledge through an illustration that depicts comparative routes comprising genetic/environmental/immune variables leading to those distinctive disease phenotypes.

5. Table 2: Please provide the references of the findings. Also, clarify the symbols with labels.

6. Table 3: Please provide the percentage estimates and sample size of the clinical features positive for SNRA and SPRA along with references.

7. Table 4: Again, please provide references for the claimed differences in response to therapy. 

Author Response

Responses to Reviewer #2

(Responses to reviewers' comments have been highlighted in green throughout the text)

Paroli and Sirinian have reviewed the less prevalent and less severe nonetheless less clear seronegative aspects of rheumatoid arthritis (SNRA) than its seropositive counterpart (SPRA). The authors highlighted on the knowledge base of genetic and immune risk factors for pathogenesis, current diagnosis and therapeutic aspects of SNRA. The key insights include dysregulation of inflammatory signaling pathway (e.g., PD-1) as a possible mechanism leading to SNRA, miRNA expression and SNP in genes as possible molecular biomarkers, and the need to reassess the current RA clinical diagnosis criteria to classify SNRA. The topics of discussion in this review are merited however some clarification into the authors’ insights is needed.

  1. Please summarize in a table the distinctive features of SNRA and SPRA including but not limited to epidemiology, clinical features, biomarkers, diagnosis, therapeutic and prognostic features. Answer: The data on the different characteristics of SNRA versus SPRA were originally distributed in 4 tables. In light of reviewer's suggestion to add a figure representing the distinguishing characteristics of SNRA and SPRA, we have included that information in this figure instead of an additional table to make it easier to read the available evidence on the topic

  1. Table 1: It is not clear what positive and negative mean in the context since as I understand the results are based on miRNA/mRNA expression data. Please modify the table to clarify the fold change ratio in the expression. Also, add the references to those experimental findings in the table. Answer: We have modified the table by replacing the + and - symbols with more understandable definitions. However, quantitative data and percentages on gene expression are very difficult to obtain from current literature. Indeed, we speak of odd ratios or increases or reduced expression often assessed qualitatively by molecular biology techniques or by specific laboratory measurement systems. However, we have added as suggested the most relevant references related to the different points indicated in the table
  2. Pathogenesis –

paragraph 1: It is interesting to note the association of non-HLA genes with SNRA but not SPRA. Could the authors also highlight the functional role of the genes (and if any other) within the context of RA? Answer: The possible role of mutations in non-HLA genes in the pathogenesis of SNRA, and in particular the correlation with innate immunity, was added in the text.

paragraph 5: If SpA is mentioned here for the first time, please write the full form. Answer: The abbreviation SpA has been carefully checked and corrected accordingly

  1. It would be easier for a layperson reader to follow the contributing factors to the onset and progression of SNRA and SPRA based on available knowledge through an illustration that depicts comparative routes comprising genetic/environmental/immune variables leading to those distinctive disease phenotypes. Answer: As mentioned earlier, we have added a figure (Figure 1) to illustrate more clearly the most relevant differences between SNRA and SPRA
  2. Table 2: Please provide the references of the findings. Also, clarify the symbols with labels. Answer: Symbols have been replaced by labels and relevant bibliographic references have been added.
  3. Table 3: Please provide the percentage estimates and sample size of the clinical features positive for SNRA and SPRA along with references. Answer: Percentages and sample size related to SNRA were not quantitatively studied in detail. Therefore, qualitative data are reported
  4. Table 4: Again, please provide references for the claimed differences in response to therapy. Answer: The bibliography related to the data shown in Table 4 has been added.

Round 2

Reviewer 2 Report

The revised version of the manuscript by Paroni and Srinian has been well-supplemented with illustration and references. I only have a minor comment:

1. Illustration: Remove numbering (1, 2, 3, 4...) and put factor titles upfront (e.g., Synovitis: Rare) throughout. Maintain center alignment of the text in all boxes.

Author Response

Response to reviewer#2

1. Illustration: Remove numbering (1, 2, 3, 4...) and put factor titles upfront (e.g., Synovitis: Rare) throughout. Maintain center alignment of the text in all boxes. Answer: in Figure 1, the numbers have been removed, the factors titles have been placed up front, and the centering of the text within all boxes has been maintained.